# Multi-Granularity Aggregation with Spatiotemporal Consistency for Video-Based Person Re-Identification

**DOI:** 10.3390/s24072229

**Published:** 2024-03-30

**Authors:** Hean Sung Lee, Minjung Kim, Sungjun Jang, Han Byeol Bae, Sangyoun Lee

**Affiliations:** 1School of Electrical and Electronic Engineering, Yonsei University, 50 Yonsei-ro, Seodaemun-gu, Seoul 03722, Republic of Korea; hslee2860@yonsei.ac.kr (H.S.L.); mjkima@yonsei.ac.kr (M.K.); jeu2250@yonsei.ac.kr (S.J.); 2School of Computer Science and Engineering, Kunsan National University, 558 Daehak-ro, Gunsan-si 54150, Republic of Korea; hbbae@kunsan.ac.kr

**Keywords:** video-based person re-identification, spatiotemporal learning, attention mechanism, complementary learning

## Abstract

Video-based person re-identification (ReID) aims to exploit relevant features from spatial and temporal knowledge. Widely used methods include the part- and attention-based approaches for suppressing irrelevant spatial–temporal features. However, it is still challenging to overcome inconsistencies across video frames due to occlusion and imperfect detection. These mismatches make temporal processing ineffective and create an imbalance of crucial spatial information. To address these problems, we propose the Spatiotemporal Multi-Granularity Aggregation (ST-MGA) method, which is specifically designed to accumulate relevant features with spatiotemporally consistent cues. The proposed framework consists of three main stages: extraction, which extracts spatiotemporally consistent partial information; augmentation, which augments the partial information with different granularity levels; and aggregation, which effectively aggregates the augmented spatiotemporal information. We first introduce the consistent part-attention (CPA) module, which extracts spatiotemporally consistent and well-aligned attentive parts. Sub-parts derived from CPA provide temporally consistent semantic information, solving misalignment problems in videos due to occlusion or inaccurate detection, and maximize the efficiency of aggregation through uniform partial information. To enhance the diversity of spatial and temporal cues, we introduce the Multi-Attention Part Augmentation (MA-PA) block, which incorporates fine parts at various granular levels, and the Long-/Short-term Temporal Augmentation (LS-TA) block, designed to capture both long- and short-term temporal relations. Using densely separated part cues, ST-MGA fully exploits and aggregates the spatiotemporal multi-granular patterns by comparing relations between parts and scales. In the experiments, the proposed ST-MGA renders state-of-the-art performance on several video-based ReID benchmarks (i.e., MARS, DukeMTMC-VideoReID, and LS-VID).

## 1. Introduction

Person re-identification (ReID) is an essential application in large-scale surveillance systems and smart cities, aiming to identify individuals across different times and locations amidst varying conditions (e.g., camera views, occlusion, background clutter, illumination, scale, and body pose). With the growth of video surveillance systems, advanced video ReID methods [1,2,3,4,5,6,7,8] have been attracting attention due to their potential to offer larger capacity for achieving more robust performance. In contrast to image-based ReID, which relies solely on a single image, video-based approaches harness a richer source of temporal information. Consequently, most video-based methods [6,7,8,9,10,11,12,13,14] predominantly concentrate on feature extraction and aggregating such spatiotemporal knowledge.

Previous methods commonly fall into two categories: part-based [14,15,16,17] and attention-based approaches [4,7,11,13,18,19,20]. These methods segment global features into partial information and then derive a single feature vector by leveraging relationships among relevant spatial and temporal knowledge. Specifically, the methods [14,15] spatially separate images or global features into fixed partitions. Ref. [14] employed horizontal separation and utilized the graph convolutional network (GCN) [21] to aggregate spatial and temporal dimensions. Similarly, Ref. [15] employed horizontal separation at multiple scales to divide details and aggregate them through hypergraphs at various granularity levels. Despite these efforts, challenges persist due to temporal misalignment caused by object occlusion or inaccurate detection algorithms during feature aggregation. When features are separated into horizontal parts, they inconsistently include unnecessary information along the temporal axis during occlusion or detection errors (Figure 1a). Such inconsistencies, particularly in video ReID, potentially lead to interference in features and result in inaccurate outcomes. Alternatively, attention mechanisms such as [22,23,24,25] have been widely utilized to enhance feature representation by accentuating relevant regions while suppressing irrelevant areas. Recent video-based ReID approaches [18,19] have explored attention-based partitioning methods to leverage diverse attention parts. However, these methods typically create sub-attentions separate from the fixed main attention, resulting in imbalanced information across parts and restricting the number of semantic part divisions. They may tend to prioritize parts with abundant information, potentially overlooking finer details of targets with relatively lesser information. This could result in inaccurate outcomes when crucial parts of the target are occluded. To overcome the above problem, we aim to extract enhanced ReID features by fully exploiting detailed information in the spatiotemporal information by ensuring uniform information quantity across parts and maintaining consistent semantic information temporally.

In this paper, we introduce the *consistent part-attention* (CPA) module, which effectively manages uniform spatiotemporal information without interference or noise. Notably, CPA learns uniform attention in spatiotemporal dimensions solely through self-information and a few priors, eliminating the need for hard labels such as human parsing or skeleton data. As illustrated in Figure 1b, the CPA module not only eliminates interference and noise in spacetime, but also ensures consistent delivery of semantic information to the model, averting uneven information distribution and ensuring the thorough capture of fine target details.

Addressing the challenge of video ReID entails leveraging both spatial and temporal knowledge across various granularities. To this end, we employed the *Multi-Attentive Part Augmentation* (MA-PA) scheme to obtain multi-granularity information with various attention scales. Multi-granularity information [15,26] has shown promise in incorporating detailed features in videos. The MA-PA generates multi-granularity attention by recombining fine attention from CPA. As shown in Figure 2a, merging segmented part information can alter and diversify semantic meanings (i.e., when the target is small, distinguishing facial or footwear details at a smaller scale becomes challenging; however, combining these with parts related to shirts or pants extends semantic information to upper and lower body regions). This empowers the model to capture robust information across a spectrum of semantic meanings, from fine-grained to broader details.

To capture temporal relations, we employed the *Long-/Short-term Temporal-Augmentation* (LS-TA) module, which obtains multi-granularity temporal information. LS-TA conducts time sampling at different intervals to harness the overall temporal advantage. Long- and short-term temporal cues have been utilized for temporal modeling due to their respective crucial patterns [19,27,28]. As shown in Figure 2b, varying sampling intervals yield distinct features. For instance, short-term clues reflect the target’s motion patterns, while long-term cues effectively alleviate occlusion. Consequently, LS-TA yields diverse temporal features, enabling the model to extract robust features in various situations. After augmenting spatial and temporal granular cues, we propose the *Spatiotemporal Multi-Granularity Aggregation* (ST-MGA) to exploit densely separated spatial and temporal clues simultaneously. ST-MGA investigates the relations between multi-granular and part information from both spatial and temporal cues. Since the granular part features refined in the previous process contain all the information in spatiotemporal dimensions without interference, ST-MGA can extract robust and complementary features in any situation.

To summarize, our main contributions are as follows. We designed a *consistent part-attention* (CPA) module to provide spatiotemporally consistent and well-aligned part attention. To exploit the multi-scale granularity, we introduce *Multi-Attention Part Augmentation* (MA-PA), which uses the fine part features from CPA to synthesize semantic parts at multiple scales spatially. We also suggest *Long-/Short-term Temporal Augmentation* (LS-TA) for considering relations at various temporal scales. The temporally consistent part information through CPA allows LS-TA to have the full advantage of temporal knowledge. Using the spatiotemporal multi-granularity part information, we propose *Spatiotemporal Multi-Granularity Aggregation* (ST-MGA), which performs partwise and scalewise aggregation. The ST-MGA method investigates the relations between multi-granular and part information from both spatial and temporal cues and encourages complementary features for video person ReID. In the experiments, we validated the effectiveness of our approach on multiple benchmarks. Our approach outperforms previous state-of-the-art methods on several video ReID benchmarks and shows more accurate attention parts than the existing part-based approaches.

## 2. Related Work

### 2.1. Video-Based Person ReID

In recent years, video-based person ReID [7,14,15,17,18,19,20,26,27,28,29,30,31,32,33,34,35,36,37,38,39,40,41,42,43] has garnered significant attention due to the abundant temporal and spatial cues available in videos. The predominant approach in video ReID is extracting and aggregating dynamic spatiotemporal features. Some methods employ recurrent architectures [5,6,44] for video representation learning to leverage temporal cues. Refs. [28,45] utilized 3D-convolution [46,47] for spatial–temporal feature learning. A temporal attention mechanism [8,9,20,48] has also been proposed for robust temporal feature aggregation. In recent research, to contain richer temporal and spatial information, many methods [20,39,40,41,42] have been proposed. Ref. [39] presented a statistic attention (SA) block to capture long-range high-order dependencies of the feature maps. Ref. [40] used hierarchical mining, which mines the characteristics of pedestrians by referring to the temporal and intra-class knowledge. Ref. [41] proposed a saliency and granularity mining network to learn the temporally invariant features. Ref. [42] implemented a two-branch architecture to separately learn the pose feature and appearance feature and concatenated them together for more discriminative representation. Ref. [20] removed interference and obtained key pixels and frames by learning attention-guided interference-removal modules. A simple literature survey of the previous methods is shown in Table 1.

Recently, refs. [7,26] focused on aggregating diverse partial information, both spatially and temporally. To obtain partial spatial cues, certain approaches have adopted horizontal partitioning [14,15,16,17] or explored diverse attention mechanism [7,11,18,19]. However, most of these methods cannot fully exploit the potential of spatiotemporal knowledge. Horizontal partitioning often struggles to maintain information consistency in cases of temporal misalignment due to an inaccurate detector. The diverse attention mechanisms have an unbalanced information distribution regarding attention, leading to inefficient aggregation. To exploit the full advantage of spatiotemporal information, we first propose a straightforward, yet effective framework, called *consistent part attention* (CPA), designed to ensure the consistency of partial information and lead to efficient aggregation in the spatial and temporal dimensions. Then, we efficiently the aggregate spatiotemporal partial information using the *Spatiotemporal Multi-Granularity Aggregation* (ST-MGA) scheme to extract complementary video features.

### 2.2. Attention for Person ReID

The attention mechanism, as discussed in [22,23,24,25], has been widely used in person ReID to enhance representation by emphasizing the relevant features and suppressing irrelevant ones. In image-based ReID, Refs. [49,50,51,52,53,54,55,56,57,58] learned attention in terms of the spatial or channel dimension. In some studies of video-based ReID, temporal attention is performed to weigh and aggregate frame-level features [5,9,59]. Moreover, Ref. [26] proposed joint spatial and temporal attention to exploit relations at multiple granularities. Recently, Refs. [18,19] proposed diverse spatial-attention modules to enhance video representation. The diverse attention modules focus on different regions for consecutive frames. However, they create sub-attention parts separate from the attention of the main frame, which restricts the number of semantic attention parts and results in each part containing an inconsistent amount of information. This limitation leads to inefficiencies in aggregating diverse features because the focus remains on the main attention part. Unlike the above methods, the proposed CPA provides uniform spatial information and temporally coincident cues about the diverse attention parts, leading to efficient aggregation in the spatial and temporal dimensions.

### 2.3. Spatiotemporal Aggregation

Capturing spatial and temporal information is critical to learning comprehensive representations of videos effectively. The most-used approach [4,18,27,32,34] involves using convolutional neural networks (CNNs) to extract spatial features from individual video frames and integrating these features with temporal modeling. Various methods, such as RNNs, 3D-CNNs, GCNs, and attention mechanisms, can be employed for spatiotemporal aggregation. Ref. [28] introduced a compact 3D convolutional kernel that facilitates multi-scale temporal feature learning by incorporating long-term temporal modeling and refining appearance features through spatiotemporal masking. Ref. [19] focused on capturing visual features by considering the spatial details and long-distance context information, which are combined using a multi-scale temporal kernel in the 3D convolutional layers. Ref. [60] proposed graph convolution to directly propagate cross-spacetime and cross-scale information, capturing high-order spatial–temporal correlations. Ref. [35] addressed the problem of spatial distractors by memorizing them and suppressing distracting scene details while using temporal attention patterns to aggregate the frame-level representation. Ref. [61] learned spatiotemporal information attention using ConvLSTM [62] to explicitly capture and aggregate spatiotemporal information in video-based industrial smoke emission recognition. Ref. [63] explored spatial correlations within each frame to determine the attention weight of different locations and also considered temporal correlations between adjacent frames.

## 3. Methodologies

In this section, we propose *Spatiotemporal Multi-Granularity Aggregation* (ST-MGA) methods with the proposed spatiotemporally consistent cues. We introduce the preliminaries in Section 3.1. Then, we describe the proposed *consistent part-attention* (CPA) module, which aims to provide spatiotemporally consistent information by using a simple, but effective attention approach, in Section 3.2. For multi-granular spatial and temporal information, we introduce *Multi-Attention Part Augmentation* (MA-PA) and *Long-/Short-term Temporal Augmentation* in Section 3.3. Last, we present ST-MGA for spatiotemporally complementary features in Section 3.4.

### 3.1. Overview

To improve video-based person ReID, we aimed to extract consistent spatial and temporal cues, and aggregated this information for complementary feature extraction. The overall framework of the proposed approach is illustrated in Figure 3. As the input video clip, we randomly sampled *T* frames as $V=\{I_{1},I_{2},\cdots ,I_{T}\}$. Then, we extracted video features $F\in R^{T\times C\times H\times W}$ through the CNN backbone (e.g., ResNet-50 [64] pretrained on ImageNet [65]), where *H*, *W*, and *C* represent the height, width, and number of channels, respectively. To extract the part cues, we employed the CPA module, which ensures that extracted attentions contain consistent spatiotemporal semantic information. Using CPA, we obtained $N_{s}$ sub-attentions, $A=\{a^{n}|n=1,2,\cdots ,N_{s}\}$. Subsequently, we extracted multi-granular parts using the MA-PA method. MA-PA augments multi-granular attention with varying scales by utilizing attention parts from CPA. Augmented multi-granular attention was applied to *F* to generate $N_{P}$ granular features. To consider temporal relations, we also jointly augmented $N_{t}$ long- and short-term features using LS-TA by combining multi-scale temporal cues. We define part features after MA-PA and LS-TA as $\hat{P}$ and $\tilde{P}$, respectively. Leveraging $N_{p}\times N_{t}$ multi-granular features, we propose ST-MGA, which exploits spatial and temporal relations and aggregates different levels of part features. Finally, after partial and temporal averaging of the aggregated features, we extracted the complementary video features, $Z_{c}\in R^{C}$. The notations and their corresponding descriptions are presented in Table 2.

### 3.2. Spatiotemporally Consistent Part Attention

Given the *t*-th global features $F_{t}\in R^{H\times W\times C}$, we first optimized the global attention through a learnable model. We extracted the global attention value $a_{t}^{g}$ for the *t*-th frame from the global-attention module GA composed as follows: (1)$$\begin{array}{c} a_{t}^{g}=Sigmoid\left(W_{2}ReLU\left(W_{1}F_{t}\right)\right), \end{array}$$
where $W_{1}\in R^{\frac{C}{\gamma }\times C}$ and $W_{2}\in R^{1\times \frac{C}{\gamma }}$ are implemented by 1×1 convolution with shrink ratio γ followed by BN. Subsequently, we can extract global attentive features $P_{g}\in R^{C}$ as: (2)$$\begin{array}{c} P_{g}=\frac{1}{T}\sum_{t=1}^{T}\left(P\left(F_{t}+a_{t}^{g}⊙F_{t}\right)\right), \end{array}$$
where the symbol ⊙ represents elementwise multiplication and P is the global max-pooling operation. To optimize $a^{g}$, we applied the batch hard triplet loss [66] and softmax cross-entropy loss with $P_{g}$ as the input. The two loss formulas are denoted as $L_{tri}^{g}$ and $L_{CE}^{g}$.

After obtaining $a^{g}$, we optimized the CPA by using $a^{g}$. As illustrated in Figure 4, CPA composes $N_{s}$ part-attention modules PA, which extract $N_{s}$ different semantic sub-attentions $A_{t}={\left\{a_{t}^{n}\right\}}_{n=1}^{N_{s}}$. Each PA module has the same structure as GA, with different parameters. To encourage each attention focus to have different semantic information, we define the following priors: (i) the features applied by each attention must be semantically classifiable for each other; (ii) the sum of all sub-attention parts should be global attention; (iii) each sub-attention part should contain a uniform amount of information.

Based on the first prior, we designed a part classifier that classified $N_{s}$ parts as the class ID. We first extracted the part-attention features $P={\left\{P_{n}\right\}}_{n=1}^{N_{s}}$ in the same way as $P_{g}$ with the corresponding attention. Then, we trained each attentive feature to be classifiable by setting a different label $y^{n}$, a one-hot vector for the *n*-th attentive part. To this end, we employed the softmax cross-entropy loss with input $P_{n}$. We define this loss formula as follows: (3)$$\begin{array}{c} L_{PCE}=E\left[-log\left(p\left(y^{n}|P_{n}\right)\right)\right], \end{array}$$
where $p(y^{n}|P_{n})$ is the predicted probability that input $P_{n}$ belongs to its part label $y^{n}$. $L_{PCE}$ encourages each attention to focus on different relevant areas because each attentive feature must be semantically classifiable despite being applied to the same input features.

Before applying the second and third priors, we introduced a technique to extract precise part-attention information. The technique is that sub-attentions only learn the residual from the background value of the global attention when directly inducing each attention part. The rationale behind this approach is to focus on the relevant region only, rather than the background. If background information is not excluded, arbitrary attention is slightly highlighted on the background region as the interest, making interference during aggregation. The goal is for each attention part to have equal importance; thus, we proceeded with learning by extracting only the residuals from the background values of the global attention. By excluding the background information, attention is directed to the relevant areas only. We analyze this in Section 4.

As shown in Figure 4, we first define the minimum of $a^{g}$ of the background score and obtained the residual global attention, ${\dot{a}}^{g}$, by removing it. We define the residual part attentions as $\dot{A}={\left\{{\dot{a}}^{n}\right\}}_{n=1}^{N_{s}}$, which are learned by focusing only on the semantic area and ignoring unnecessary parts (e.g., the background and occlusions). We utilized $\dot{a}$ and ${\dot{a}}^{g}$ to impose constraints on the second and third priors. First, we designed the L1 distance loss, which sets the sum of all residual part attentions $\dot{A}$ equal to ${\dot{a}}^{g}$. Additionally, we encouraged the spatial sum of each residual part attention to match the spatial sum of ${\tilde{a}}^{g}$ divided by $N_{s}$. We combined the first and second loss formulas as follows: (4)$$\begin{array}{c} L_{PA}=\left|\right|\sum_{n=1}^{N_{s}}{\dot{a}}^{n}-{\dot{a}}^{g}\left|\right|+\sum_{n=1}^{N_{s}}\left|\right|\sum_{k=1}^{K}{\dot{a}}_{k}^{n}-\frac{1}{N_{s}}\sum_{k=1}^{K}{\dot{a}}_{k}^{g}\left|\right|, \end{array}$$
where K=H×W. $L_{PA}$ simply, but strongly satisfies the prior we predefined. Furthermore, $L_{PA}$ encourages an emphasis on the position of relevant and uniform importance in each part.

The overall loss function for training CPA is formulated as follows:
(5)$$\begin{array}{c} L_{CPA}=L_{tri}^{g}+L_{CE}^{g}+L_{PA}+\lambda L_{PCE}, \end{array}$$
where λ is a scaling parameter for weighting $L_{PCE}$. The attention parts extracted using CPA represent temporally consistent semantic cues with uniform spatial importance. We leveraged these attention parts instead of the previous simple horizontal partitions to take full advantage of this in the video. In subsequent sessions, we will denote $\tilde{A}$ as *A* for convenience.

### 3.3. Multi-Granularity Feature Augmentation

To further improve video feature representation, we augmented diverse spatial and temporal cues at multi-granular levels. With the efficient CPA module to obtain consistent part attention *A*, we applied Multi-Attentive Part Augmentation (MA-PA), which uses *A* as a sub-attention to generate the parent attention. Following [15], we hierarchically combined the sub-attention into a granular scale m∈{0,1,2,⋯,M}. To avoid excessive duplication of certain parts, we prevented overlapping when combining sub-attentions, as shown in Figure 2a. The parent granular attention comprises the non-duplicated combination of child sub-attentions. We define the *t*-th augmented part attention as $\hat{A_{t}}=\left\{A_{t}^{m}|m=0,1,2,\cdots ,M\right\}$, where ${\hat{A}}_{t}^{m}=\left\{{\hat{a}}_{t}^{m,n}|n=1,2,\cdots ,2^{m}\right\}$. ${\hat{a}}_{t}^{m,n}$ represents attention to the *n*-th part of the *m*-th scale, and each scale has $2^{m}$ part attentions. Subsequently, we obtained $N_{P}=\sum_{m=0}^{M}2^{m}$ attentive features $\hat{P}$ in each frame as follows: (6)$$\begin{array}{c} {\hat{p}}_{t}^{m,n}=P\left(F_{t}+{\hat{a}}_{t}^{m,n}⊙F_{t}\right), \end{array}$$
where ${\hat{p}}_{t}^{m,n}$/${\hat{a}}_{t}^{m,n}$ is the *n*-th part feature/attention with scale *m* and P is the global max-pooling operation. As with $\hat{P}$, each part feature at the same granular level contains uniform semantic information.

Using augmented spatial attentive features $\hat{P}$, we employed Long-/Short-term Temporal Augmentation (LS-TA) to augment long- and short-term features jointly. As shown in Figure 2b, the importance of long- and short-term temporal relations can vary depending on the sequence context. However, simple temporal processing is often inefficient due to temporal inconsistencies such as misalignment between adjacent frames [32]. In our study, we have already addressed these temporal inconsistency problems with the proposed CPA; therefore, we can proceed with the temporal granularity aggregation without concern. To this end, we first applied temporal attention from each frame level by predicting the framewise scores as follows: (7)$$\begin{array}{c} {\tilde{p}}_{t}={\hat{p}}_{t}+a_{t}^{\prime }⊙{\hat{p}}_{t}, \end{array}$$
(8)$$\begin{array}{c} a_{t}^{\prime }=Sigmoid\left(W_{2}^{\prime }ReLU\left(W_{1}^{\prime }hayp_{t}\right)\right),\; \end{array}$$
where $a_{t}^{\prime }$ is the temporal attention value, $W_{1}^{\prime }\in R^{\frac{C}{\gamma }\times C}$ and $W_{2}^{\prime }\in R^{1\times \frac{C}{\gamma }}$ are implemented by 1D-convolution with a kernel size of 1, and γ is the shrink ratio. Then, following [19], we conducted a temporal-select operation $\left\{S^{t^{\prime }}:\hat{P}\to {\tilde{P}}_{t^{\prime }}\in R^{T\times C}\right\}$, where S is the frame-select operation of a 1D-temporal convolution [67] with a kernel size of $2t^{\prime }+1$ and $t^{\prime }\in 1,2,\cdots ,T^{\prime }$ is the temporal granular scale. We varied $t^{\prime }$ to obtain multiple temporal granular features at different intervals from short to long term. With LS-TA, we generated $N_{T}=T\times T^{\prime }$ temporal granular features in each part and combined them with the granular part, and a total of $N_{P}\times N_{T}$ spatiotemporal granular cues were stored. We leveraged these granular cues to exploit the spatiotemporal coverage in the video.

### 3.4. Spatiotemporal Multi-Granularity Aggregation

To extract optimal video person ReID features, it is important to effectively aggregate the multi-granular information from the spatial and temporal information. From this view, we propose a Multi-Granularity-Aggregation (MGA) module that fully exploits the spatiotemporal cues. The structure of MGA is illustrated in Figure 5. MGA primarily comprises partwise and scalewise aggregators. The partwise aggregator has *M* parallel branches with the same structure, where *M* is the simplified scale number. For each scale of the granularity level, we represent *N* part features as ${\tilde{p}}_{i}^{m}\in R^{N\times C}$, where i={1,⋯,N} and *N* is the simplified number of parts. The partwise relation matrix in the *m*-th scale $R^{m}\in R^{N\times N}$ can be defined as a dot-product affinity as follows: (9)$$\begin{array}{c} R_{i,j}^{m}=\frac{\exp(\varphi {\left({\tilde{p}}_{i}^{m}\right)}^{T}\omega \left({\tilde{p}}_{j}^{m}\right)/\tau )}{\sum_{k=1}^{N}\exp\left(\varphi {\left({\tilde{p}}_{i}^{m}\right)}^{T}\omega \left({\tilde{p}}_{k}^{m}\right)/\tau \right)}, \end{array}$$
where φ and ω are linear functions and τ is the temperature hyperparameter. With the relation matrix $R^{m}$, we calculated partwise aggregated features as $\phi {\left(\tilde{p}\right)}^{T}R_{m}$, where ϕ is a linear function. Then, the output features were extracted using average pooling of all part cues, followed by the residual addition. After the partwise aggregator, we concatenated *M* outputs, denoted as $Y\in R^{M\times C}$.

With input *Y*, scalewise aggregation proceeds using a scalewise aggregator. To aggregate all granular scale cues on *Y*, we designed a scale-attention (SA) module that predicts the scalewise attention score to weigh the aggregation. We normalized the learned attention scores from the SA module via the softmax function across scale dimensions and obtained the scale-attention score $A_{s}\in R^{C\times M}$: (10)$$\begin{array}{c} A_{s}=SA\left(Y\right)=Softmax\left(W_{\theta_{2}}\left(ReLU\left(W_{\theta_{1}}Y\right)\right)/\tau \right), \end{array}$$
where $W_{\theta_{1}}\in R^{\frac{C}{\eta }\times C}$ and $W_{\theta_{2}}\in R^{1\times \frac{C}{\eta }}$ are a fully connected layer with shrink ratio η and τ is the temperature hyperparameter. Afterward, we extracted complementary video representation $Z\in R^{C}$ using scalewise attention $A_{s}$ to aggregate all scale features as follows: (11)$$\begin{array}{c} Z=\sum_{m=1}^{M}A_{s}⊙Y+A\left(Y\right). \end{array}$$

We applied MGA separately in the spatial (S-MGA) and temporal (T-MGA) information. S-MGA aggregates partwise and scalewise based on spatially attentive-part features, whereas T-MGA is based on the temporal-part features. As illustrated in Figure 5, T-MGA has the same structure as S-MGA, with *N* and *M* replaced by *T* and $T^{\prime }$, respectively. By applying T-MGA and S-MGA, we extracted final video features $Z^{C}$, enhanced by exploiting the spatiotemporal cues across each scale.

For the reliable learning of each stage, we used three types of features $Z^{\prime }=\left\{Z^{F},Z^{T},Z^{C}\right\}$, where $Z^{F}$ is the global features of the backbone, $Z^{T}$ represents the temporally aggregated features from T-MGA, and $Z^{C}$ denotes the final complementary video features from S-MGA. The overall joint objective is defined as follows: (12)$$\begin{array}{c} L_{total}=\sum_{z\in Z^{\prime }}\left(L_{CE}^{z}+L_{tri}^{z}\right)+L_{CPA}, \end{array}$$
where $L_{CE}$ is the softmax cross-entropy loss and $L_{tri}$ is the batch-hard triplet loss [66]. In the inference phase, we only used the complementary video person ReID features, $Z_{C}$.

## 4. Experiments

In this section, we extensively evaluate the proposed framework in video-based person ReID scenarios. We conducted comprehensive experiments on three benchmark datasets, and provide a detailed analysis of the results. Additionally, we included extensive ablation studies to investigate the effectiveness of our approach further.

### 4.1. Datasets and Evaluation Metric

We evaluated the proposed framework on three challenging video ReID datasets: MARS [2], DukeMTMC-VideoReID (Duke-V) [68], and LS-VID [27]. MARS is one of the large-scale benchmark datasets for video ReID, which contains 17,503 tracklets of 1261 identities and an additional distractor of 3248 tracklets. There are substantial bounding box misalignment problems to make it more challenging. Duke-V is a widely used large-scale video ReID dataset captured by 8 cameras with 4832 tracklets of 1404 identities. LS-VID is the most recent large-scale benchmark dataset for video ReID. It contains 3772 identities and 14,943 tracklets captured by 15 cameras. There are many challenging elements, such as varying illumination and bounding box misalignment, to make it close to a real-world environment, A summary comparison is illustrated in Table 3. For the evaluation, we used only the final complemented video ReID features $Z_{C}$ during the inference stage. Moreover, we assessed the performance using Rank-1, Rank-5, and Rank-10 accuracy for cumulative matching characteristics (CMCs) and the mean average precision (mAP). Rank-k and the mAP are the most popular evaluation metrics for person ReID. Rank-k measures the accuracy by evaluating whether the correct match appears within the top-k-ranked results. To this end, for each query, an algorithm will rank all the gallery samples according to their distances. The mAP evaluates how well the system ranks the retrieved matches for each query. It considers both precision and recall, providing a comprehensive assessment of the re-identification system’s performance across different query scenarios.

### 4.2. Implementation Details

Following the common practices in ReID [19,27,32,69], we used ResNet-50 [64] trained by ImageNet classification [65] as the backbone of the proposed model for a fair and effectiveness validation. Similar to [69,70], we removed the last down-sampling operation to enrich granularity, resulting in a total down-sampling ratio of 16. We employed a random sampling strategy to select 8 frames for each video sequence with a stride of 4 frames. Each batch contains 16 identities, each with 4 tracklets. The input frames were resized to 256×128. Random horizontal flip and random erasing [71] were used for data augmentation. The training process utilizes the Adam [72] optimizer for 150 epochs. The learning rate was initialized to $3.5\times 10^{-4}$ and decayed every 40 epochs using a decay factor of 0.1. For the hyperparameters of the proposed framework, a greedy search was conducted about λ and τ to increase reproducibility and transparency. λ is the weight for $L_{PCE}$ of Equation (5), and τ is the temperature hyperparameter, which indicates sensitivity to the relation of Equations (9) and (10). As shown in Figure 6, we set λ to 0.1 and τ to 0.05 as the optimal parameters. The shrink ratios γ and η were set to 16 and 4, respectively. The framework was implemented with *PyTorch1.4* [73] on one GeForce RTX 3090 (Nvidia, Santa Clara, CA, USA).

### 4.3. Ablation Study

#### 4.3.1. The Influence of CPA

We first evaluated the effect of the proposed CPA module with different components of loss functions and techniques. To extract spatiotemporally consistent information, we suggest three priors and approaches with $L_{PCE}$, $L_{PA}$, and a residual technique (res). We conducted only spatial aggregation with three granular-part attentions (1, 2, and 4 partitions). $L_{PCE}$ encourages the attentions from CPA to focus on different sub-areas. In Table 4, employing $L_{PCE}$ brings 0.3%/0.8% mAP/Rank-1 gains, which is the same as the part-based approach (in the second row in Table 5). Thus, employing $L_{PCE}$ can only successfully separate the attention parts and lead to aggregation between separate part information. $L_{PA}$ performs a comparison with the global attention and spatially unifies the amount of information in each attention, which allows separate attentions to focus on the more semantic areas, making all attention parts useful in the aggregation. However, the comparison with the global attention inevitably includes some scores for the background; thus, $L_{PA}$ alone does not ensure that the attention parts focus on the fully semantic area. To address this problem, learning only the residuals excluding the background scores (res) results in a 1.6%/1.8% mAP/Rank-1 performance increment in the last row of Table 4. The improvement proves that CPA is effective.

For a more detailed analysis of CPA, we provide a few contextual comparisons to visualize how the methods within CPA affect the actual attention map. A comparison of the attention map visualizations for CPA is presented in Figure 7. Using only $L_{PCE}$ to learn to be classifiable for each attention (2nd row) results in partitions separated by a constant amount, like the horizontal partition (1st row). However, it concurrently generates significant unnecessary background scores. When $L_{PA}$ is added without the residual technique, although it contains information about the same part with a different situation or identity, certain specific parts still retain unnecessary information, such as the background or occlusion area. This observation indicates that some parts have learned to carry attention scores for irrelevant information from global attention. When the complete CPA framework is applied, the attention remains consistent across all situations, effectively suppressing extraneous information. This shows that CPA eliminates occlusions and misalignments, providing consistent attention regardless of various contexts.

#### 4.3.2. Comparison of Part-Based and CPA Models

We conducted experiments to compare the proposed CPA framework with the previous part-based approach. With the same MGA module, we only separated the components of different partitioning methods and the number of parts. The results are presented in Table 5. Compared with the baseline, the CPA model with 4 partitions achieved a 1.5%/2.3% mAP/Rank-1 increment on MARS. Compared with the part-based approach with the same 4 partitions, the model achieved a 1.2%/1.5% mAP/Rank-1 performance increment. The model achieved the best performance for 8 partitions of CPA with a 1.5%/2.5% mAP/Rank-1 increment compared to the baseline. All CPA partitions achieved more remarkable performance than previous horizontal part-based approaches. This result indicates that the proposed CPA method is more informative than the simple horizontal partition by suppressing unnecessary background information and representing only relevant regions. As shown in Figure 8, depending on the number of parts, each attention part uniformly focuses on the most salient semantic part (e.g., Figure 8c, each attention focuses on a semantic part, such as the head, shirt, pants, and shoe). However, if attention parts are separated too much, each part has very little information, making it difficult to determine the semantic area, as depicted in the third image in Figure 8d. The superiority of the proposed CPA approach over the part-based method is observable in the first and last rows in Figure 7.

#### 4.3.3. The Influence of Granularity

To assess the effectiveness of MA-PA and LS-TA as a function of granularity, we compared different combinations of granularity scales in the spatial and temporal dimensions. Table 6 shows the details. First, following [15], we experimented with spatially scaling parts by a factor of 2. The Rank-1 performance increases steadily when more detailed spatial granularities are captured. The scaling factors (i.e., 1, 2, 4, 8) resulted in the highest Rank-1 performance with a 2.8% increment compared to the baseline, showing that the MA-PA approach is effective even with deep granularity. This indicates that the model is stronger in feature extraction when semantic information is diversified. Next, we experimented with different granular factors over temporal cues. We observed that the performance saturates when using the optimal granular in the spatial (i.e., 1, 2, 4) and temporal (i.e., 1, 3, 5) domains, respectively. We achieved a performance increment of 2.0%/2.9% mAP/Rank-1. This indicates that the model supplemented it using useful information when various sampling information was delivered over time.

#### 4.3.4. Effectiveness of ST-MGA

In ST-MGA, we aggregated the partial features in the spatial and temporal domains. First, the partwise aggregator was used to aggregate across part features, and the scalewise aggregator combines across the granularity scales. We first compared different components of the MGA to validate the partwise and scalewise aggregators. As listed in Table 7, we validate different MGA components on the presence or absence of part-specific aggregators (‘part’) and scale-specific aggregators (‘scale’). A 0.9%/0.5% mAP/Rank-1 performance increment occurred when using only the partwise aggregator, and a 0.7%/0.4% mAP/Rank-1 performance increment occurred when using only the scalewise aggregator. Combining both improved the 0.9%/1.5% mAP/Rank-1 performance, indicating that the proposed combination is more effective than simply averaging the partial features (1st row).

To further validate ST-MGA, we conducted comparative experiments on the influence of each MGA in the spatial (S-MGA) and temporal (T-MGA) domains. Table 8 presents the details. Compared with the baseline, S-MGA achieved a 1.7%/1.8% mAP/Rank-1 increment on MARS. Compared to a simple horizontal partition (P-MGA), S-MGA performed better with a 1.0%/0.4% mAP/Rank-1 increment. Combining T-MGA and P-MGA showed little performance difference compared with T-MGA. The reason may be that T-MGA does not work well for simple horizontal parts due to problems such as temporal misalignment. In contrast, when T-MGA and S-MGA were used together, a 1.1%/1.2% mAP/Rank-1 performance increment occurred compared to only using T-MGA. This result is because S-MGA only deals with semantic information through CPA, so the effect of T-MGA is complementary. To verify which order is better between T-MGA and S-MGA, we made comparisons of the above two different orders. For the Rank-1 metric only, T-MGA→S-MGA outperformed by 0.5% compared to using S-MGA→T-MGA. As a result, it was verified that the proposed model was effective by exploring knowledge in the spatial and temporal domains, regardless of order, and we used T-MGA→S-MGA with relatively high Rank-1 performance as the optimal model.

### 4.4. Comparison and Visualization

#### 4.4.1. Comparison with the State of the Art

As presented in Table 9, we compared ST-MGA with state-of-the-art methods on the MARS, Duke-V, and LS-VID datasets. For the MARS [2] dataset, ST-MGA reached 87.2% mAP and 92.0% Rank-1 results, which is the best mAP and Rank-5 accuracy, and achieved the second-best Rank-1 results. For the Duke-V [68] dataset, the best mAP and Rank-5 accuracy were achieved at 96.8% and 99.9%, respectively. For the LS-VID [27] dataset, the results were competitive, with the third-best mAP and Rank-1 at 79.3% and 88.5%, respectively. We noticed that ST-MGA outperformed ST-GCN [14] and MGH [15], which use a horizontal part-based approach. Specifically, the proposed method also outperformed TCLNet [18] and BiCnet-TKS [42], which usesimilar diverse attention-based methods, with an improvement of up to 1.4%/2.2% and 1.2%/1.8% mAP/Rank-1 accuracy in MARS, respectively. Further, ST-MGA outperformed several recent models (i.e., SINet [37], CAVIT [38], HMN [40], SGMN [41], and BIC+LGCN [42]). In particular, the proposed method shows higher accuracy than the complex transformer-based method [38], which has recently attracted attention. The above results verify the effectiveness and superiority of ST-MGA in video ReID.

#### 4.4.2. Visualization Analysis

In this section, we visualize some retrieval results from MARS [2] in Figure 9. As can be observed, it is difficult for the baseline model to accurately distinguish people who have a similar appearance or interference when there are an inaccurate detection, occlusion, and uncertain pose. In these cases, the baseline model is not properly focused on the semantic area, and includes unnecessary information about interferences and noises, resulting in relatively low Rank-1 accuracy. The ST-MGA, on the other hand, focused on semantic areas in each part and aggregated them, resulting in high Rank-1 accuracy, even when exposed to relatively difficult and complex interference and noise.

Moreover, we visualize part attention using the proposed CPA module in Figure 10. As observed, the attention parts from CPA focus on similar semantic areas, regardless of the differences between datasets. They each focus on different semantic parts of the face, top, bottom, and shoe and uniformly provide part cues. In occlusion situations, such as in Duke-V, we explored only the correct semantic area, excluding the occluded part. In particular, by focusing on the same semantic part, we can eliminate unnecessary interference in the temporal domain. This allows ST-MGA to contribute to complementary feature extraction by minimizing the interference in both the spatial and temporal domains.

## 5. Conclusions

This paper presents the ST-MGA framework, a novel approach for robust video-based person ReID. ST-MGA effectively captures consistent spatiotemporal information to mitigate interference and enhances feature extraction through comprehensive utilization of diverse spatiotemporal granular information. To tackle interference arising from spatiotemporal inconsistency, we introduce the CPA module, which learns from self-information and specific priors. The CPA module efficiently separates attention parts to extract features with spatially uniform amounts and temporally identical semantic information. Additionally, our approach employs multi-granularity feature augmentation to synthesize granular information encompassing semantic attention parts across various scales. Spatially, MA-PA extracts various semantic granular information by synthesizing fine attention without overlapping. Temporally, LS-TA augmented various granular features through various time sampling intervals. Leveraging granular information with different scales, the MGA module effectively utilizes spatiotemporal cues to extract complementary features. Within MGA, we explored the relationships between part/scale information, aggregating them based on relation scores. The resulting aggregated representations enable complementary feature extraction by prioritizing pertinent semantic information while filtering out unnecessary interference or noise. Extensive experiments corroborated the effectiveness and superiority of our ST-MGA, highlighting its potential for advancing video-based person ReID research.

## Figures and Tables

**Figure 1 sensors-24-02229-f001:**
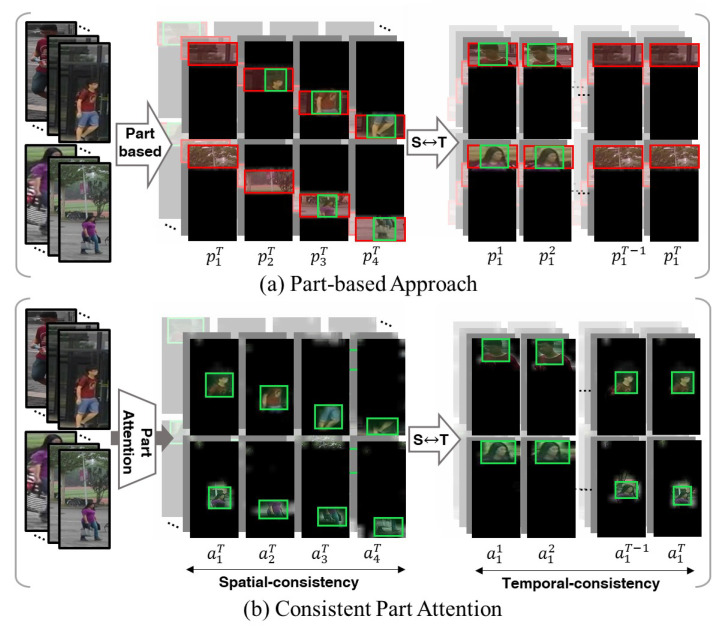
Comparison of the (**a**) part-based approach and (**b**) proposed consistent part-attention (CPA) method. As shown in the figure, the part-based approach has a different amount of relevant spatial information with some interferences for each part, whereas the CPA method provides only key information uniformly. When temporal misalignment occurs, the part-based approach demonstrates inconsistent human parts. In contrast, CPA consistently offers semantic information about the same part.

**Figure 2 sensors-24-02229-f002:**
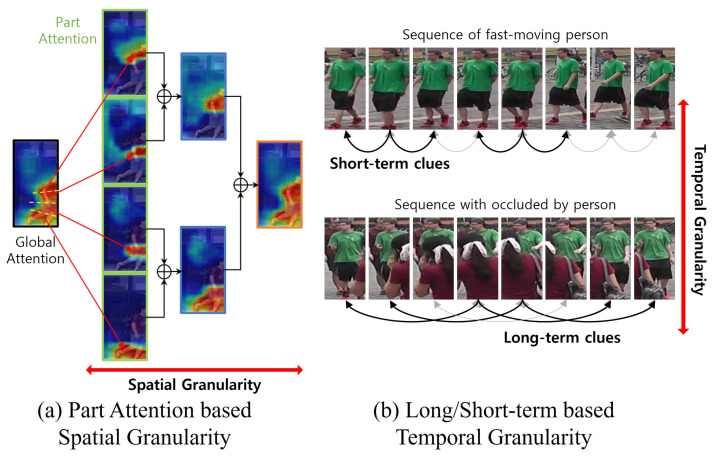
Illustration of spatiotemporal multi-granularity. (**a**) Part-attention-based spatial granularity synthesizes diverse semantic attention at multiple granular levels by combining sub-part attentions. (**b**) Long-/short-term-based temporal granularity comprises granularities with varying sampling intervals to adopt the context of different sequences for the long and short terms.

**Figure 3 sensors-24-02229-f003:**
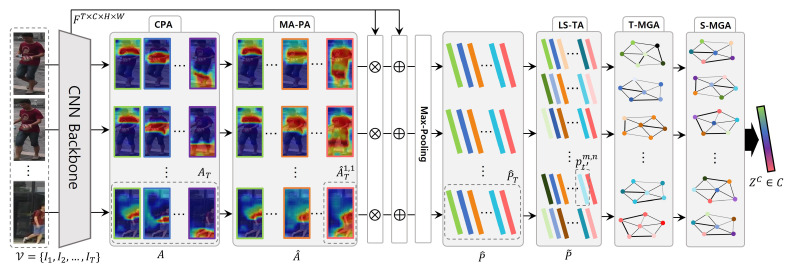
Overview of our framework. We sampled *T* frames in the video sequence and extracted features *F* from the CNN backbone. We first extracted sub-attentions *A* through the CPA module with spatiotemporally consistent information. For multiple spatial granularities, we augmented part attention $\hat{A}$ by combing *A* hierarchically. Subsequently, we extracted the part-attentive features $\hat{P}$ by applying $\hat{A}$ and the max-pooling operation to *F*. We jointly augmented $\hat{P}$ to $\tilde{P}$, which contain long-/short-term features, to consider temporal relations. Leveraging the augmented-spatiotemporal-part features, we exploited all video cues by aggregating the part features spatially (S-MGA) and temporally (T-MGA). Last, we extracted the complementary video person ReID features.

**Figure 4 sensors-24-02229-f004:**
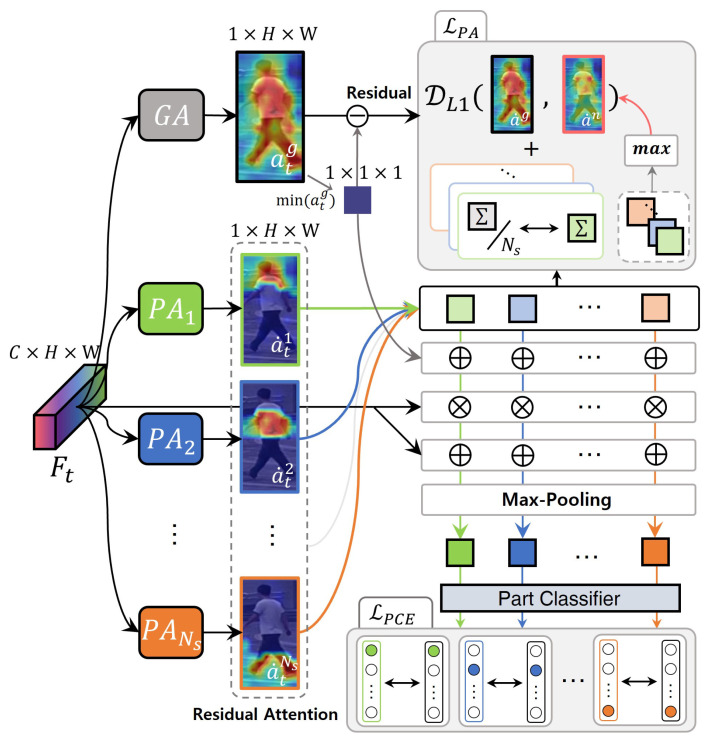
Learning framework for the consistent part-attention (CPA) module. The CPA includes one global-attention module, GA, to extract global attention $a^{g}$, $N_{s}$ part-attention modules, PA, and one part classifier, $C_{P}$. In CPA, we used $a^{g}$ to encourage each attention part to be optimized based on predefined priors.

**Figure 5 sensors-24-02229-f005:**
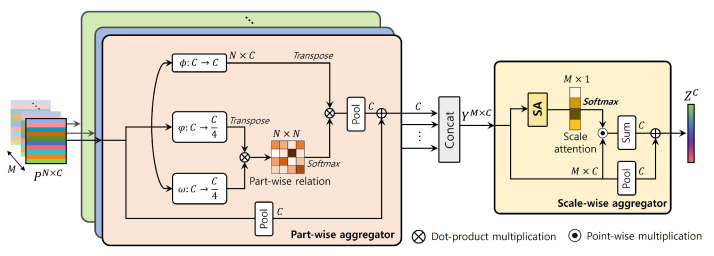
The architecture of Multi-Granularity Aggregation (MGA). As the input $\tilde{P}$, partwise aggregation proceeds first at each granular scale. Then, the granular features *X* are aggregated using a scale-attention block. Last, MGA extracts complementary video ReID features, *Z*. Temporally, *N* and *M* are changed to *T* and $T^{\prime }$, respectively.

**Figure 6 sensors-24-02229-f006:**
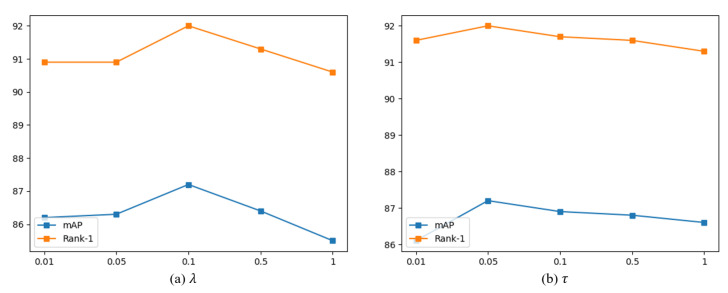
Sensitivity analysis on hyperparameters for (**a**) λ and (**b**) τ in MARS [2].

**Figure 7 sensors-24-02229-f007:**
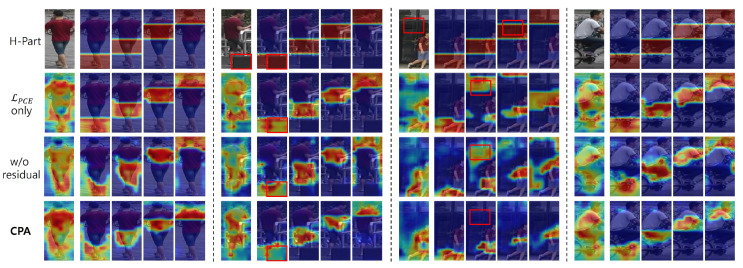
Comparison of attention map visualizations for consistent part-attention (CPA) analysis between different situations (e.g., general, occlusion, misaligned, and pose variations). When applying the full CPA, each situation displays a proper and uniformly divided attention map.

**Figure 8 sensors-24-02229-f008:**
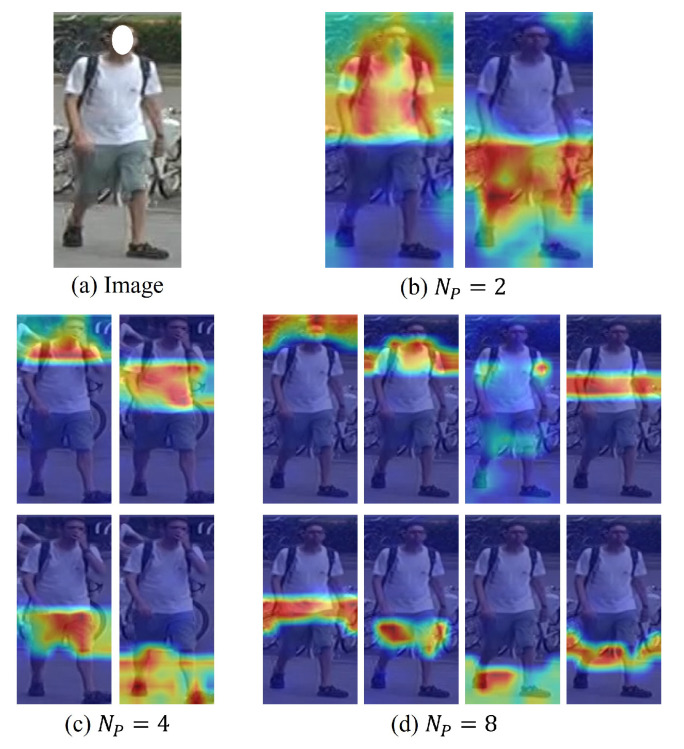
Visualization of part attention from the consistent part-attention (CPA) module with a different number of partitions $N_{p}$.

**Figure 9 sensors-24-02229-f009:**
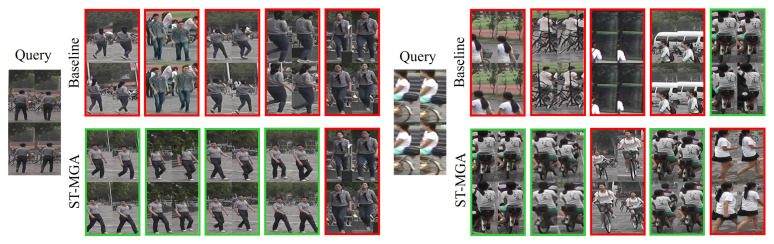
The Visualization of ReID retrieval results using the baseline and the proposed ST-MGA on MARS. For each row, the first sequence is the query, while the five sequences in the middle correspond to the Rank-1 to Rank-5 of the baseline model, and the rest are the retrieval results of our ST-MGA. The correct and incorrect matches are marked with green and red bounding boxes, respectively.

**Figure 10 sensors-24-02229-f010:**
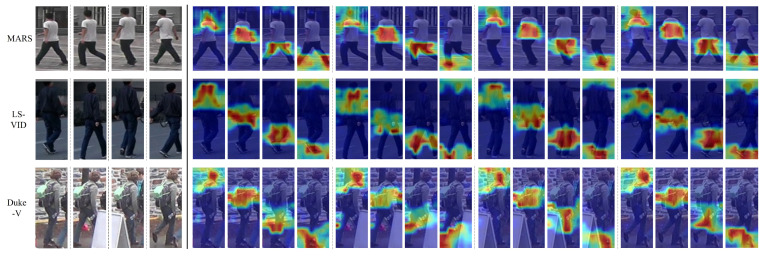
Visualization of part attentions from the consistent part-attention (CPA) module on MARS, LS-VID, and Duke-V. Showing the original image and four separated part attentions for each image for different datasets.

**Table 1 sensors-24-02229-t001:** Methodology of video-based person re-identification.

Reference	Source	Methodology
M3D [28]	AAAI’19	Proposed multi-scale 3D-convolution layer to refine the temporal features
STA [7]	AAAI’19	Proposed spatial–temporal attention approach to fully exploit discriminative parts
GLTR [27]	ICCV’19	Proposed global–local temporal representation to exploit multi-scale temporal cues
RGSA [29]	AAAI’20	Designed relation-guided spatial-attention module to explore discriminative regions
FGRA [30]	AAAI’20	Proposed frame-guided region-aligned model to extract well-aligned part features
MG-RAFA [26]	CVPR’20	Suggested attentive feature aggregation with multi-granularity information
PhD [31]	CVPR’20	Proposed Pompeiu–Hausdorff distance learning to alleviate the data noise problem
STGCN [14]	CVPR’20	Jointly optimized two GCN branches in spatial and temporal dimensions for complementary information
MGH [15]	CVPR’20	Designed a multi-granular hypergraph structure to increase representational capacities
TCLNet [18]	ECCV’20	Introduced a temporal-saliency-erasing module to focus on diverse part information
AP3D [32]	ECCV’20	Proposed appearance-preserving 3D-convolution to align the adjacent features at the pixel level
AFA [33]	ECCV’20	Proposed adversarial feature augmentation, which highlights the temporal coherence features
SSN3D [34]	AAAI’21	Designed a self-separated network to seek out the same parts in different frames
BiCnet-TKS [19]	CVPR’21	Used multiple parallel and diverse attention modules to discover diverse body parts
STMN [35]	ICCV’21	Leveraged spatial and temporal memories to refine frame-/sequence-level representations
STRF [36]	ICCV’21	Proposed spatiotemporal representation factorization for learning discriminative features
SINet [37]	CVPR’22	Designed SINet to enlarge attention regions for consecutive frames gradually
CAVIT [38]	ECCV’22	Used a contextual alignment vision transformer for spatiotemporal interaction
SANet [39]	TCSVT’22	Introduced the SA block, which can capture long-range and high-order information
HMN [40]	TCSVT’22	Designed hierarchical mining network which can mine as many characteristics
SGMN [41]	TCSVT’22	Designed a saliency- and granularity-mining network for discovering temporal coherence
BIC+LGCN [42]	TCSVT’23	Used a branch architecture to separately learn appearance features and human pose
IRNet-V [20]	TCSVT’23	Proposed an interference-removal framework for removing various interferences

**Table 2 sensors-24-02229-t002:** Notations and descriptions.

Notations	Descriptions
V	Input video
*T*	Length of V
*F*	Output feature from backbone
*A*	Set of part attentions from CPA
$\dot{A}$	Set of residual part attentions in CPA
*P*	Set of part-attentive features
$\hat{A}$	Set of augmented part attentions from MA-PA
$\hat{P}$	Set of augmented spatial attentive features after MA-PA
$a^{\prime }$	Temporal attention value in LS-TA
$\tilde{P}$	Set of augmented multi-granular features after LS-TA
$Z^{F}$	Global averaged feature from backbone
$Z^{T}$	Temporal aggregated feature from T-MGA
$Z^{C}$	Final complementary video feature from S-MGA

**Table 3 sensors-24-02229-t003:** Statistics between different video ReID datasets.

	MARS	Duke-V	LS-VID
# Identities	1261	1404	3772
# of Videos	20,751	4832	14,943
# of Cameras	6	8	15
B-Box	DPM	manual	FRCNN

**Table 4 sensors-24-02229-t004:** Performance of consistent part-attention (CPA) module in MARS [2] under different loss components. **Bold** denotes best performance.

$L_{\mathrm{PCE}}$	$L_{\mathrm{PA}}$	res	mAP	Rank-1	Rank-5	Rank-10
			85.2	89.1	86.7	97.5
✓			85.5	89.9	96.9	97.7
✓	✓		85.6	90.1	96.9	97.9
✓	✓	✓	**86.8**	**90.9**	**97.4**	**98.0**

**Table 5 sensors-24-02229-t005:** Comparison between part-based method and consistent part attention (CPA) in MARS [2]. **Bold** denotes best performance.

Model	mAP	Rank-1	Rank-5	Rank-10
Baseline	85.2	89.1	86.7	97.5
Part 4	85.5	89.9	96.9	97.7
Part 8	85.6	90.1	96.9	97.9
CPA 2	86.5	90.7	97.0	97.8
CPA 4	**86.7**	91.4	97.0	97.9
CPA 8	**86.7**	**91.6**	**97.1**	**98.1**

**Table 6 sensors-24-02229-t006:** Comparison between ‘spatial’ and ‘temporal’ scaling factors under different granularity parts in MARS [2]. **Bold** denotes best performance.

Spatial	Temporal	mAP	Rank-1	Rank-5	Rank-10
-	-	85.2	89.1	86.7	97.5
1, 2	1	86.5	90.7	97.0	97.8
1, 2, 4	1	**86.7**	91.1	**97.5**	**98.0**
1, 2, 4, 8	1	**86.7**	**91.9**	97.2	97.8
1, 2, 4	1, 3	86.8	90.6	97.2	97.9
1, 2, 4	1, 3, 5	**87.2**	**92.0**	97.3	98.1
1, 2, 4, 8	1, 3	87.0	90.9	**97.4**	**98.1**
1, 2, 4, 8	1, 3, 5	86.9	91.2	**97.4**	98.0

**Table 7 sensors-24-02229-t007:** Performance of ST-MGA in MARS [2] on the presence or absence of partwise aggregator (**‘Part’**) and scalewise aggregator (**‘Scale’**) in MGA architecture. **Bold** denotes best performance.

Part	Scale	mAP	Rank-1	Rank-5	Rank-10
		86.3	90.5	97.1	97.8
✓		**87.2**	91.0	**97.3**	**98.1**
	✓	87.0	90.9	**97.3**	**98.1**
✓	✓	**87.2**	**92.0**	**97.3**	98.0

**Table 8 sensors-24-02229-t008:** Performance of ST-MGA in MARS [2] under different spatial and temporal combinations. **Bold** denotes best performance.

Model	mAP	Rank-1	Rank-5	Rank-10
Baseline	85.2	89.1	86.7	97.5
P-MGA	85.9	90.5	96.5	97.6
S-MGA	86.9	90.9	97.2	**98.1**
T-MGA	86.1	90.8	97.0	97.8
T-MGA →P-MGA	86.0	90.9	96.9	97.8
S-MGA →T-MGA	**87.2**	91.5	**97.3**	**98.1**
T-MGA →S-MGA	**87.2**	**92.0**	**97.3**	98.0

**Table 9 sensors-24-02229-t009:** Quantitative comparison with state-of-the-art methods. **Red** denotes best performance, and **Blue** and **green** denote the second- and third-best performance, respectively.

Method	MARS			Duke-V			LS-VID		
	mAP	Rank1	Rank5	mAP	Rank-1	Rank-5	mAP	Rank-1	Rank-5
M3D [28]	74.1	84.4	-	-	-	-	40.1	57.7	-
STA [7]	80.8	86.3	95.7	94.9	96.2	**99.3**	-	-	-
GLTR [27]	78.5	87.0	95.8	93.7	96.3	**99.3**	44.3	63.1	**77.2**
RGSA [29]	84.0	89.4	-	95.8	97.2	-	-	-	-
FGRA [30]	81.2	87.3	96.0	-	-	-	-	-	-
MG-RAFA [26]	85.8	90.0	86.7	-	-	-	-	-	-
PhD [31]	85.8	90.0	96.7	-	-	-	-	-	-
STGCN [14]	83.7	90.0	86.4	95.7	97.3	**99.3**	-	-	-
MGH [15]	85.8	90.0	96.7	-	-	-	-	-	-
TCLNet [18]	85.8	89.8	-	96.2	96.9	-	-	-	-
AP3D [32]	85.1	90.1	-	95.6	96.3	-	-	-	
AFA [33]	82.9	90.2	96.6	95.4	97.2	**99.4**	-	-	-
SSN3D [34]	86.2	90.1	96.6	96.3	96.8	98.8	-	-	-
BiCnet-TKS [19]	86.0	90.2	-	96.1	96.3	-	75.1	84.6	-
STMN [35]	84.5	90.5	-	95.9	97.0		69.2	82.1	-
STRF [36]	86.1	90.3	-	96.4	**97.4**	-	-	-	-
SINet [37]	86.2	91.0	-	-	-	-	**79.6**	87.4	-
CAVIT [38]	**87.2**	90.8	-	-	-	-	79.2	**89.2**	-
SANet [39]	86.0	**91.2**	**97.1**	**96.7**	**97.7**	**99.9**	-	-	-
HMN [40]	82.6	88.5	96.2	96.1	96.3	-	-	-	-
SGMN [41]	85.4	90.8	-	96.3	96.9	-	-	-	-
BIC+LGCN [42]	**86.5**	91.1	**97.2**	**96.5**	97.1	98.8	-	-	-
IRNet-V [20]	**87.0**	**92.5**	-	-	-	-	**80.5**	**89.4**	-
**ST-MGA** (ours)	**87.2**	**92.0**	**97.3**	**96.8**	**97.6**	**99.9**	**79.3**	**88.5**	**96.1**

## Data Availability

No new data were created or analyzed in this study. Data sharing is not applicable to this article.

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
