# Peer review of "Multi-Granularity Aggregation with Spatiotemporal Consistency for Video-Based Person Re-Identification"

_sensors, 2024, doi:10.3390/s24072229_

Round 1

Reviewer 1 Report

Comments and Suggestions for Authors

The authors propose a spatiotemporal multi-granularity aggregation (ST-MGA) method
which is specifically designed to accumulate relevant features with spatiotemporal consistent cues. Here are few minor modifications required before acceptance:
1) A separate section for Literature survey should be added where the discussion of the existing model should be discussed.
2) A tabluar format representation of the existing literatures should be also added.
3) A clarification is required why the authors have specifically chosen ResNet-50 for ImageNet classification at the backbone of the proposed model?
4) Very recent literatures should be added as a reference.
5) The conclusion section must be improved

Comments on the Quality of English Language

English language and style are fine/minor spell check required

Author Response

Thank you for the meticulous review, which allowed us to make up for what we missed and helped to improve our paper substantially. Our paper has been corrected and revised in response to your concerns. The changes made to the manuscript are in blue-colored text. In the paper, the parts that have been revised either larger or smaller are indicated in red-colored text in Revision_v2.pdf.

1) A separate section for Literature survey should be added where the discussion of the existing model should be discussed.

Thank you for your observations and comments. For the recent literature survey on video-based ReID, we added a subsection for 2.1. Video-based person ReID. For more detailed literature survey, the existing contents were supplemented as follows:

#page 4

“2.1. Video-based Person ReID

In recent years, video-based person ReID [7, 14, 15 ,17–20, 26–43] has garnered significant attention due to the abundant temporal and spatial cues available in videos. The predominant approach in video ReID is extracting and aggregating dynamic spatiotemporal features. Some methods employ recurrent architectures [5, 6 ,44] for video representation learning to leverage temporal cues. [28, 45] utilized 3D-convolution [46, 47] for spatiotemporal feature learning. A temporal attention mechanism [ 8, 9, 48] has also been proposed for robust temporal feature aggregation. In recent research, to contain richer temporal and spatial information, many methods [20 ,39 –42] have been proposed. [39] presented a statistic attention (SA) block to capture long-range high-order dependencies of the feature maps. [40] used hierarchical mining which mines characteristics of pedestrians by referring to the temporal and intra-class knowledge. [41] proposed saliency and granularity mining network to learn the temporally invariant features. [42] implemented a two-branch architecture to separately learn the pose feature and appearance feature, and concatenate them together for more discriminative representation. [20] removed interference and got key pixels and frames by learning attention-guided interference removal modules.

Recently, [7, 26] have focused on aggregating diverse partial information both spatially and temporally. To obtain partial spatial cues, certain approaches have adopted horizontal partitioning [14– 17] or explore diverse attention mechanism [7, 11 , 18, 19]. However, most of these methods can not fully exploit the potential of spatiotemporal knowledge. Horizontal partitioning often struggles to maintain information consistency in cases of temporal misalignment due to an inaccurate detector. …”

2) A tabluar format representation of the existing literatures should be also added.

As suggested, a summary of existing literatures related to person video-based re-identification was briefly added to Table 1 in page 4.

These additions aim to provide readers with a comprehensive understanding of the current state-of-the-art in video-based person ReID research.

3) A clarification is required why the authors have specifically chosen ResNet-50 for ImageNet classification at the backbone of the proposed model?

The use of a backbone based on ResNet50 is a common practice of CNN-based models in the ReID task. We used ResNet50 in the same way for fair and effective verification, and the contents were added to the Section4.2 with some references, as follows:

#Page 10

“4.2. Implementation Details

Following the common practices in ReID [19, 27, 32, 69], we used ResNet-50 [64] trained by ImageNet classification [65] as the backbone of the proposed model for fair and effectiveness validation.

4) Very recent literatures should be added as a reference.

We have included five recent video-based person ReID literatures as below.

[39] Bai, Shutao, et al. "SANet: Statistic attention network for video-based person re-identification." IEEE Transactions on Circuits and Systems for Video Technology 32.6 (2021): 3866-3879.

[40] Wang, Zhikang, et al. "Robust video-based person re-identification by hierarchical mining." IEEE Transactions on Circuits and Systems for Video Technology 32.12 (2021): 8179-8191.

[41] Chen, Cuiqun, et al. "Saliency and granularity: Discovering temporal coherence for video-based person re-identification." IEEE Transactions on Circuits and Systems for Video Technology 32.9 (2022): 6100-6112.

[42] Pan, Honghu, et al. "Pose-aided video-based person re-identification via recurrent graph convolutional network." IEEE Transactions on Circuits and Systems for Video Technology (2023).

[20] Tao, Huanjie, Qianyue Duan, and Jianfeng An. "An adaptive interference removal framework for video person re-identification." IEEE Transactions on Circuits and Systems for Video Technology (2023).

The contents of these papers have been incorporated into the related-works section.

We also reflected the performance comparison in Table 9. Even when compared to the most recent state-of-the-art, our proposed method demonstrates competitive performance.

5) The conclusion section must be improved

In order to improve the conclusion, the contents were revised as follows:

#Page 15

“This paper presents the Multi-Granularity Aggregation method with spatiotemporal consistent cues to address video-based person ReID. The proposed consistent part attention module provides spatiotemporal consistent and well-aligned part attention. The part attention loss and part cross-entropy loss encourage spatiotemporally consistent partial attention extraction. The multi-attention part aggregation module spatially synthesizes semantic attention parts at multiple scales, and the long- and short-term temporal augmentation module generates long- and short-term cues using the temporal kernel with different strides. With multi-granular spatial and temporal information, the multi-granular aggregation module exploits spatiotemporal cues to encourage complementary features in video ReID. We first investigated the relations between the partial information and aggregated them with the part-wise relation scores. Then, we performed scale-wise aggregation with the scale-attention module. Aggregating spatiotemporally consistent and granular parts enables complementary and robust feature extraction by investigating only useful semantic information while excluding unnecessary interference or noise. Extensive experiments demonstrate the effectiveness of the proposed method.”

-> “This paper presents the ST-MGA framework, a novel approach for robust video-based person ReID. ST-MGA effectively captures consistent spatiotemporal information to mitigate interference and enhances feature extraction through the comprehensive utilization of diverse spatiotemporal granular information. To tackle interference arising from spatiotemporal inconsistency, we introduce the CPA module, which learns from self-information and specific priors. The CPA module efficiently separates attention parts to extract features with spatially uniform amounts and temporally identical semantic information. Additionally, our approach employs multi-granularity feature augmentation to synthesize granular information encompassing semantic attention parts across various scales. Spatially, MA-PA extracts various semantic granular information by synthesizing fine attention without overlapping. Temporally, LS-TA augmented various granular features through various time sampling intervals. Leveraging granular information with different scales, the MGA module effectively utilizes spatiotemporal cues to extract complementary features. Within MGA, we explore the relationships between part/scale information, aggregating them based on relation scores. The resulting aggregated representations enable complementary feature extraction by prioritizing pertinent semantic information while filtering out unnecessary interference or noise. Extensive experiments corroborate the effectiveness and superiority of our ST-MGA, highlighting its potential for advancing video-based person ReID research.”

Reviewer 2 Report

Comments and Suggestions for Authors

Video-based person Re-Identification (ReID) is an intriguing topic, yet persistent challenges arise from inconsistencies across video frames, stemming from occlusion and imperfect detection. These factors hinder both temporal processing and the balance of spatial information. This paper aims to address these challenges by accumulating relevant features with consistent spatiotemporal cues. The consistent part attention (CPA) module extracts well-aligned attentive parts, effectively resolving misalignment issues in videos and enhancing aggregation efficiency. The experiments demonstrate positive results, confirming the advantages of the proposed scheme.

Several improvements are suggested and listed below:

    1. 1. The notations should undergo a thorough review to ensure consistency and clarity. It's essential to define all notations clearly to prevent any confusion.

    2.  
    3. 2. Consider listing the parameters used in the experiments to enhance reproducibility and transparency.

    4.  
    5. 3. Clearly define measured metrics such as mAP (mean Average Precision), Rank1 to Rank10 for better understanding and interpretation.

    6.  
    7. 4. Define 'Part' and 'Scale' in Table 5 to provide clear context and understanding.

    8.  
    9. 5. In line 353, the sentence "For the Rank1 metric only, T-MGA -> S-MGA outperformed by 0.5%" should be rewritten to improve clarity and precision.

    10.  
    11. 6. Check the entire article for grammar errors to ensure clarity and readability.

  1.  

  2.  
  3.  

Comments on the Quality of English Language

Please check for grammar errors throughout the entire article to ensure clarity and readability.

Author Response

Thank you for the meticulous review, which allowed us to make up for what we missed and helped to improve our paper substantially. Our paper has been corrected and revised in response to your concerns. The changes made to the manuscript are in blue-colored text. In the paper, the parts that have been revised either larger or smaller are indicated in red-colored text in Revision_v2.pdf.

1) The notations should undergo a thorough review to ensure consistency and clarity. It's essential to define all notations clearly to prevent any confusion

Thank you for your observations and comments. We have identified confusion regarding the notations we used, and have made revisions accordingly. Additionally, we have streamlined the notation within the overall framework diagram for clarity and added "Table.2 Notations and Descriptions" to ensure the clarity of notations, as follows:

2) Consider listing the parameters used in the experiments to enhance reproducibility and transparency.

To enhance reproducibility and transparency, experiments were conducted regarding the parameters based on the MARS dataset. A grid search was performed on the lambda, which represents the weight for PCE loss, and the temperature hyperparameter tau for the part/scale relation. The results of this are included in Figure 6. Additionally, the following content has been added to Section 4.2 to reflect experiments.

Figure 6. Sensitivity analysis on hyperparameters for (a) λ and (b) τ in MARS [2]

#Page 11

“4.2. Implementation Details

Following the common practices in ReID [19,27,32,69], we used ResNet-50 [64] trained by ImageNet classification [65] as the backbone of the proposed model for fair and effectiveness validation. Similar to [69,70], we removed the last down-sampling operation to enrich granularity, resulting in a total down-sampling ratio of 16. We employed a random sampling strategy to select 8 frames for each video sequence with a stride of four frames. Each batch contains 16 identities, each with 4 tracklets. For the hyperparameters of the proposed framework, a greedy search was conducted about λ and τ to increase reproducibility and transparency. λ is weight for LPCE of Eq. 5 and τ is temperature hyperparameter that indicates sensitivity to relation of Eq. 9 and Eq. 10. As shown in Fig. 6, we set λ to 0.1 and τ to 0.05 as the optimal parameter.

3) Clearly define measured metrics such as mAP (mean Average Precision), Rank1 to Rank10 for better understanding and interpretation.

Thank you for the helpful advice. In order to facilitate better understanding for our readers, we have provided detailed explanations of the metrics we used as follows:

#Page 10

“4.1. Datasets and Evaluation Metric

We evaluate the proposed framework on three challenging video ReID datasets:, MARS [2], DukeMTMC-VideoReID (Duke-V) [68] and LS-VID [27].

There are many challenging elements, such as varying illumination and bounding box misalignment, to make it close to a real-world environment, A summary comparison is illustrated in Table 1. For evaluation, We used only the final complemented video ReID features ZC during the inference stage. Moreover, we assessed the performance using Rank-1, Rank-5 and Rank-10 accuracy for cumulative matching characteristics (CMC) and the mean average precision (mAP). The Rank-k and mAP are most popular evaluation metrics for person ReID. Rank-k measures the accuracy by evaluating whether the correct match appears within the top-k ranked results. To this end, for each query, an algorithm will rank all the gallery samples according to their distances. evaluates how well the system ranks the retrieved matches for each query. It considers both precision and recall, providing a comprehensive assessment of the re-identification system’s performance across different query scenarios.

4) Define 'Part' and 'Scale' in Table 5 to provide clear context and understanding.

We apologize for any confusion caused by the lack of clarity. In response, we have provided additional information regarding 'Part' and 'Scale' in Section 4.3.4 as follows:

#Page 13

“4.2.4. Effectiveness of ST-MGA

In ST-MGA, we aggregated the partial features in spatial and temporal. First, the part-wise aggregator is used to aggregate across part features, and the scale-wise aggregator combines across the granularity scales. As listed in Table 5, we validate different MGA components on the presence or absence of part-specific aggregators (’Part’) and scale-specific aggregators (’Scale’).

Additionally, we have ensured clear delineation of the explanations for 'Part' and 'Scale' within the description of Table 5:

“Table 5. Performance of ST-MGA in MARS [2] on the presence or absence of part-wise aggregator (’Part’) and scale-wise aggregator (’Scale’) in MGA architecture”

5) In line 353, the sentence "For the Rank1 metric only, T-MGA -> S-MGA outperformed by 0.5%" should be rewritten to improve clarity and precision

We apologize for the unclear expression in that sentence. In order to improve clarity and precision, we have added the following explanation in Section 4.2.4.

#Page 14

“4.2.4. Effectiveness of ST-MGA

To further validate ST-MGA, we conducted comparative experiments on the influence of each MGA in spatial (S-MGA) and temporal (T-MGA). Table 6 presents the details. Compared with the baseline, S-MGA achieved a 1.7%/1.8% mAP/Rank1 increment on MARS. Compared to a simple horizontal partition (P-MGA), S-MGA performed better with a 1.0%/0.4% mAP/Rank1 increment. Combining T-MGA and P-MGA showed little performance difference compared with the T-MGA. The reason may be that T-MGA does not work well for simple horizontal parts due to such problems as temporal misalignment. In contrast, when T-MGA and S-MGA were used together, a 1.1%/1.2% mAP/Rank1 performance increment occurred compared to only using T-MGA. This result is because S-MGA only deals with semantic information through CPA, so the effect of T-MGA is complementary. To verify which order is better between T-MGA and S-MGA, we make of the above two different orders. For the Rank1 metric only, T-MGA → S-MGA outperformed by 0.5% compared to using S-MGA → T-MGA . As a result, it was verified that the proposed model was effective by exploring knowledge in spatial and temporal, regardless of order, and we used T-MGA → S-MGA with relatively high rank-1 performance as the optimal model.

6) Check the entire article for grammar errors to ensure clarity and readability.

We apologize for not accurately expressing our English style and grammar. We further revised and corrected the English and context of the entire paper and reflected your comments.

Reviewer 3 Report

Comments and Suggestions for Authors

The manuscript proposes the spatiotemporal multi-granularity aggregation (ST-MGA) method which is specifically designed to accumulate relevant features with spatiotemporal consistent cues. The proposed framework consists of three main stages: Extraction, which extracts spatiotemporally consistent partial information; Augmentation, which augments the partial information with different granularity levels; and Aggregation, which effectively aggregates the augmented spatiotemporal information.

This article needs major revision, and the authors should address the following issues:

1) It would be beneficial to provide a clearer overview of the challenges in person ReID and a more explicit statement of the specific problems or limitations addressed by the proposed method.

2) I suggest the authors further emphasize the novelty or innovation brought by each module, while introducing the proposed modules (CPA, MA-PA, LS-TA, ST-MGA).

3) Please provide a clearer and more detailed analysis of the experimental results to demonstrate why the mentioned method in the paper is superior.

4) The paper could benefit from a careful proofreading to improve the clarity and coherence of the language. Additionally, consider refining sentence structures for better readability.

5) The paper would benefit from a discussion of more recent works. Please review the following relevant works and discuss them in the article: (1) An adaptive interference removal framework for video person Re-Identification. (2) Attention-aggregated Attribute-aware Network with Redundancy Reduction Convolution for Video-based Industrial Smoke Emission Recognition.

Author Response

Thank you for the meticulous review, which allowed us to make up for what we missed and helped to improve our paper substantially. Our paper has been corrected and revised in response to your concerns. The changes made to the manuscript are in blue-colored text. In the paper, the parts that have been revised either larger or smaller are indicated in red-colored text in Revision_v2.pdf.

1) It would be beneficial to provide a clearer overview of the challenges in person ReID and a more explicit statement of the specific problems or limitations addressed by the proposed method.

- Thank you for your observations and comments. The problems addressed in video-based person re-identification (ReID) and the corresponding purpose in this paper are as follows.

In video-based person re-identification (ReID), the typical approach is to fully leverage spatial and temporal information. However, unnecessary interference and noise in the spatiotemporal domain make it challenging to sufficiently utilize spatial and temporal information. Particularly, part-based approaches, as illustrated in Figure 1, inconsistently include unnecessary information along the temporal axis during occlusion or detection errors, potentially leading to interference in features. And diverse attention-based methods effectively suppress unnecessary spatial information but exhibit an uneven distribution of semantic information across attention parts. They will tend to prioritize parts with abundant information, potentially overlooking finer details of targets with relatively lesser information. This could result in inaccurate outcomes when crucial parts of the target are occluded. To overcome the above problem, this paper aims to extract enhanced ReID features by fully exploiting detailed information in the spatiotemporal by ensuring uniform information quantity across parts and maintaining consistent semantic information temporally.

Reflecting on the above view, we extended this in Section 1. Introduction as follows:

#Page 2

“(Introduction to previous part-based methods) The methods [14,15] are simply spatially separate images or global features used to fixed partitions. [14] performed a horizontal separation and aggregated the spatial and temporal dimensions using the graph convolutional network (GCN) [21]. One study [15] also used horizontal separation at multiple scales to split the details, and aggregate them through hypergraphs at various granularity levels. Despite these efforts, challenges persist due to temporal misalignment caused by object occlusion or inaccurate detection algorithms during feature aggregation. When features are separated into horizontal parts, they inconsistently include unnecessary information along the temporal axis during occlusion or detection errors (Fig. 1 (a)). Such inconsistencies, particularly in video ReID, potentially lead to interference in features and result in inaccurate outcomes. Alternatively, attention mechanisms such as [22–25] have been widely utilized to enhance feature representation by accentuating relevant regions while suppressing irrelevant areas. Recent video-based ReID approaches [18,19] have explored attention-based partitioning methods to leverage diverse attention parts. However, these methods typically create sub-attentions separate from the fixed main attention, resulting in imbalanced information across parts and restricting the number of semantic part divisions. They may tend to prioritize parts with abundant information, potentially overlooking finer details of targets with relatively lesser information. This could result in inaccurate outcomes when crucial parts of the target are occluded. To overcome the above problem, this paper aims to extract enhanced ReID features by fully exploiting detailed information in the spatiotemporal by ensuring uniform information quantity across parts and maintaining consistent semantic information temporally.

2) I suggest the authors further emphasize the novelty or innovation brought by each module, while introducing the proposed modules (CPA, MA-PA, LS-TA, ST-MGA).

- As suggested, we think it is necessary to emphasize the novelty for the proposed modules.

The proposed Consistent Part Attention (CPA) module serves as a main component in managing uniform spatiotemporal information without interference or noise. CPA learns uniform attention in spatiotemporal by only utilizing self-information and a few priors, obviating the need for hard labels. This not only eliminates interference and noise in spacetime but also ensures consistent delivery of semantic information to the model, averting uneven information distribution and ensuring the thorough capture of fine target details.

The Multi-Granular Attention (MA-PA) module extracts granular information across various scales. As delineated in Figure 2 (a), merging segmented part information can alter and diversify semantic meanings. For instance, minor details such as shoes or hair may not individually yield significant features, but their amalgamation with other parts contributes to robust features, such as upper or lower body regions. This empowers the model to capture robust information across a spectrum of semantic meanings, from fine-grained to broader information.

The proposed Long-Short Term Attention (LS-TA) module conducts time sampling at different intervals to harness the overall temporal advantage. As depicted in Figure 2 (b), varying sampling intervals encourage distinct features. For instance, short-term clues reflect the target's motion patterns, while long-term clues are effective in alleviating occlusion. LS-TA yields diverse temporal features, enabling the model to extract robust features in various situations.

ST-MGA investigates the relations between multi-granular and part information from both spatial and temporal cues. Since the granular part features refined in the previous process contain all the information in spatiotemporal without interference, ST-MGA can extract robust and complementary features in any situation.

Reflecting on the above, we extended this in Section 1.Introduction as follows:

#Page 3

“In this paper, we introduce the Consistent Part Attention (CPA) module, which effectively manages uniform spatiotemporal information without interference or noise. Notably, CPA learns uniform attention in spatiotemporal dimensions solely through self-information and a few priors, eliminating the need for hard labels such as human parsing or skeleton data. As illustrated in Fig. 1 (b), the CPA module not only eliminates interference and noise in spacetime but also ensures consistent delivery of semantic information to the model, averting uneven information distribution and ensuring the thorough capture of fine target details.

Addressing the challenge of video ReID entails leveraging both spatial and temporal knowledge across various granularities. To this end, we employ the Multi-Granularity Attention (MA-PA) scheme to obtain multi-granularity information with various attention scales. Multi-granularity information [15, 26] has shown promise in incorporating detailed features in videos. The MA-PA generates multi-granularity attention by recombining fine attention from CPA. As shown in Fig. 2 (a), merging segmented part information can alter and diversify semantic meanings (for example, when the target is small, distinguishing facial or footwear details at a smaller scale becomes challenging. However, combining these with parts related to shirts or pants extends semantic information to upper and lower body regions). This empowers the model to capture robust information across a spectrum of semantic meanings, from fine-grained to broader details.

To capture temporal relations, we employ the Long-Short Term Attention (LS-TA) module, which obtains multi-granularity temporal information. LS-TA conducts time sampling at different intervals to harness the overall temporal advantage. Long- and short-term temporal cues have been utilized for temporal modeling due to their respective crucial patterns [19, 27, 28]. As shown in Fig. 2 (b), varying sampling intervals yield distinct features. For instance, short-term clues reflect the target's motion patterns, while long-term cues effectively alleviate occlusion. Consequently, LS-TA yields diverse temporal features, enabling the model to extract robust features in various situations.

After augmenting spatial and temporal granular cues, we propose the Spatiotemporal Multi-Granular Aggregation (ST-MGA) to exploit densely separated spatial and temporal clues simultaneously. ST-MGA investigates the relations between multi-granular and part information from both spatial and temporal cues. Since the granular part features refined in the previous process contain all the information in spatiotemporal dimensions without interference, ST-MGA can extract robust and complementary features in any situation.

3) Please provide a clearer and more detailed analysis of the experimental results to demonstrate why the mentioned method in the paper is superior.

- As suggested, we further complemented the analysis of the experimental results to clarify the strength of the paper.

In the case of the CPA module, the change of attention according to the different component is shown in Figure 7. When the complete CPA framework is applied, the attention remains consistent across all situations, effectively suppressing extraneous information. This shows that the CPA eliminates occlusions and misalignments, providing consistent attention regardless of various contexts. In Table 4, even using the CPA module alone, the performance improvement of mAP 1.6% and Rank1 1.8% than baseline. The improvement proves that CPA is effective. In addition, compared to the general part-based method, the performance improvement of mAP 1.2% and Rank1 1.7%, shown in Table 5. This result indicates that the proposed CPA method is more informative than the simple horizontal partition by suppressing unnecessary background information and representing only relevant regions.
In the case of MA-PA and LS-TA, the experiment was conducted according to the number of parts divided in spatial and temporal (in Table 6). When only MA-PA was conducted, the performance improvement of mAP 1.5 and Rank1 2.0 was shown, indicating that the model is stronger in feature extraction when semantic information is diversified. When LS-TA was applied, the performance improvement of mAP 0.5 and Rank1 0.9 compared to applying only MA-PA. This indicates that the model supplemented it using useful information when various sampling information was delivered over time. In Table 7, we validate different MGA components on the presence or absence of part-specific aggregators (’Part’) and scale-specific aggregators (’Scale’).

Combining both improved the 0.9%/1.5% mAP/Rank1 performance than simply averaging, indicating that the proposed combination is effective. ST-MGA has a final performance of mAP 87.2 and Rank1 92.0%, which is competitive in recent SOTA comparisons (Table 9). This results verify the effectiveness and superiority of ST-MGA in video ReID.

We extended the above view to section 4.3 ablation study.

4) The paper could benefit from a careful proofreading to improve the clarity and coherence of the language. Additionally, consider refining sentence structures for better readability.

- We apologize for not accurately expressing our English style and grammar. We further revised and corrected the English and context of the entire paper and reflected your comments.

5) The paper would benefit from a discussion of more recent works. Please review the following relevant works and discuss them in the article: (1) An adaptive interference removal framework for video person Re-Identification. (2) Attention-aggregated Attribute-aware Network with Redundancy Reduction Convolution for Video-based Industrial Smoke Emission Recognition.

- We are grateful for the benefit information. Including above works[20, 61], we have added some recent literatures [39-42] to the related works as follows:

#page 4

“2.1. Video-based Person ReID

In recent years, video-based person ReID [7, 14, 15 ,17–20, 26–43] has garnered significant attention due to the abundant temporal and spatial cues available in videos. The predominant approach in video ReID is extracting and aggregating dynamic spatiotemporal features. Some methods employ recurrent architectures [5, 6 ,44] for video representation learning to leverage temporal cues. [28, 45] utilized 3D-convolution [46, 47] for spatiotemporal feature learning. A temporal attention mechanism [ 8, 9, 48] has also been proposed for robust temporal feature aggregation. In recent research, to contain richer temporal and spatial information, many methods [20 ,39 –42] have been proposed. [39] presented a statistic attention (SA) block to capture long-range high-order dependencies of the feature maps. [40] used hierarchical mining which mines characteristics of pedestrians by referring to the temporal and intra-class knowledge. [41] proposed saliency and granularity mining network to learn the temporally invariant features. [42] implemented a two-branch architecture to separately learn the pose feature and appearance feature, and concatenate them together for more discriminative representation. [20] removed interference and got key pixels and frames by learning attention-guided interference removal modules."

#page 5

“2.3. Spatio-Temporal Aggregation

Capturing spatial and temporal information is critical to learning comprehensive representations of videos effectively. The most-used approach [4,18,27,32,34] involves using convolutional neural networks (CNNs) to extract spatial features from individual video frames and integrating these features with temporal modeling.

[35] addressed the problem of spatial distractors by memorizing them and suppressing distracting scene details while using temporal attention patterns to aggregate the frame-level representation. [61] learned spatiotemporal information attention using ConvLSTM[62] to explicitly capture and aggregate spatiotemporal information in video-based industrial smoke emission recognition. [63] explored spatial correlations within each frame to determine the attention weight of different locations, and also consider temporal correlations between adjacent frames.”

- We also the performance comparison for the newly added recent works in Table 9. Even when compared to the most recent state-of-the-art, our proposed method demonstrates competitive performance.

[39] Bai, Shutao, et al. "SANet: Statistic attention network for video-based person re-identification." IEEE Transactions on Circuits and Systems for Video Technology 32.6 (2021): 3866-3879.

[40] Wang, Zhikang, et al. "Robust video-based person re-identification by hierarchical mining." IEEE Transactions on Circuits and Systems for Video Technology 32.12 (2021): 8179-8191.

[41] Chen, Cuiqun, et al. "Saliency and granularity: Discovering temporal coherence for video-based person re-identification." IEEE Transactions on Circuits and Systems for Video Technology 32.9 (2022): 6100-6112.

[42] Pan, Honghu, et al. "Pose-aided video-based person re-identification via recurrent graph convolutional network." IEEE Transactions on Circuits and Systems for Video Technology (2023).

[20] Tao, Huanjie, Qianyue Duan, and Jianfeng An. "An adaptive interference removal framework for video person re-identification." IEEE Transactions on Circuits and Systems for Video Technology (2023).

[61] Tao, Huanjie, et al. "Attention-aggregated attribute-aware network with redundancy reduction convolution for video-based industrial smoke emission recognition." IEEE Transactions on Industrial Informatics 18.11 (2022): 7653-7664.

[62] Shi, Xingjian, et al. "Convolutional LSTM network: A machine learning approach for precipitation nowcasting." Advances in neural information processing systems 28 (2015).

Round 2

Reviewer 3 Report

Comments and Suggestions for Authors

All my concerns have been solved and I recommend publishing this work.